# Competing-Risk Analysis of Death and End Stage Kidney Disease by Hyperkalaemia Status in Non-Dialysis Chronic Kidney Disease Patients Receiving Stable Nephrology Care

**DOI:** 10.3390/jcm7120499

**Published:** 2018-12-01

**Authors:** Michele Provenzano, Roberto Minutolo, Paolo Chiodini, Vincenzo Bellizzi, Felice Nappi, Domenico Russo, Silvio Borrelli, Carlo Garofalo, Carmela Iodice, Toni De Stefano, Giuseppe Conte, Hiddo J. L. Heerspink, Luca De Nicola

**Affiliations:** 1Division of Nephrology, Department of Advanced Medical and Surgical Sciences, Nephrology Unit-University of Campania “Luigi Vanvitelli”, 80138 Naples, Italy; michiprov@hotmail.it (M.P.); Roberto.MINUTOLO@unicampania.it (R.M.); dott.silvioborrelli@gmail.com (S.B.); carlo.garofalo@hotmail.it (C.G.); carmelaiodice1821@yahoo.it (C.I.); toni-ds@live.it (T.D.S.); giuseppe.conte@unicampania.it (G.C.); 2Medical Statistics Unit, the University of Campania “Luigi Vanvitelli”, 80138 Naples, Italy; paolo.chiodini@unicampania.it; 3Nephrology Unit, Hospital Ruggi d’Aragona, 84131 Salerno, Italy; vincenzo.bellizzi@tin.it; 4Nephrology Unit, Hospital Santa Maria della Pietà, 80035 Nola, Italy; fel_nappi@alice.it; 5Nephrology Unit, Department of Public Health, University Federico II of Naples, 80131 Naples, Italy; Domenico.russo@unina.it; 6Department of Clinical Pharmacy and Pharmacology, University of Groningen, University Medical Center Groningen, 9727 Groningen, The Netherlands; h.j.lambers.heerspink@umcg.nl

**Keywords:** CKD, ESKD, death, anti-RAS, hyperkalaemia, competing risk

## Abstract

Hyperkalaemia burden in non-dialysis chronic kidney disease (CKD) under nephrology care is undefined. We prospectively followed 2443 patients with two visits (referral and control with 12-month interval) in 46 nephrology clinics. Patients were stratified in four categories of hyperkalaemia (serum potassium, sK ≥ 5.0 mEq/L) by sK at visit 1 and 2: Absent (no-no), Resolving (yes-no), New Onset (no-yes), Persistent (yes-yes). We assessed competing risks of end stage kidney disease (ESKD) and death after visit 2. Age was 65 ± 15 years, eGFR 35 ± 17 mL/min/1.73 m^2^, proteinuria 0.40 (0.14–1.21) g/24 h. In the two visits sK was 4.8 ± 0.6 and levels ≥6 mEq/L were observed in 4%. Hyperkalaemia was absent in 46%, resolving 17%, new onset 15% and persistent 22%. Renin-angiotensin-system inhibitors (RASI) were prescribed in 79% patients. During 3.6-year follow-up, 567 patients reached ESKD and 349 died. Multivariable competing risk analysis (sub-hazard ratio-sHR, 95% Confidence Interval-CI) evidenced that new onset (sHR 1.34, 95% CI 1.05–1.72) and persistent (sHR 1.27, 95% CI 1.02–1.58) hyperkalaemia predicted higher ESKD risk versus absent, independently from main determinants of outcome including eGFR change. Conversely, no effect on mortality was observed. Results were confirmed by testing sK as continuous variable. Therefore, in CKD under nephrology care, mild-to-moderate hyperkalaemia status is common (37%) and predicts per se higher ESKD risk but not mortality.

## 1. Introduction

Chronic hyperkalaemia is common in non-dialysis chronic kidney disease (ND-CKD). Recent observational studies have reported a greater prevalence of high serum potassium (sK) in this population than in general population; however, the reported rates have been extremely variable—from 1% to 50%—being mainly dependent on GFR and comorbidities of included patients, as on the number of sK measurements [1,2,3,4,5,6,7]. This latter aspect is crucial because prevalence of hyperkalaemia increases with the number of sK tests, thus indicating that identification of true hyperkalaemia cannot be based on a single sK value [6]. 

More important, an additional lack of solid information on hyperkalaemia burden in CKD is the prognostic role. Indeed, while in unselected population hyperkalaemia poses a risk excess of mortality for even moderate increases of sK (≥5.0 mEq/L) [7,8,9,10,11,12,13], the association between hyperkalaemia and progression to end-stage kidney disease (ESKD) remains controversial [1,4,10,13,14,15]. Gaining insights into this association is essential because a main determinant of hyperkalaemia is the prescription of inhibitors of renin-angiotensin-system (RASI) [16,17], that are the first-choice antihypertensive agents in CKD due to the nephroprotective efficacy [18]. In this regard, it is interesting that RASI withdrawal driven by hyperkalaemia increases mortality of patients seen in the general medicine setting [17,19]. A vicious circle may similarly ensue with hyperkalaemia onset and dependent non-use or withdrawal of RASI leading to ESKD. Increased awareness of this phenomenon and adequate management of hyperkalaemia through dietary and/or pharmacological intervention may interrupt this circle [20,21,22].

To date, no detailed study has evaluated the effect of hyperkalaemia on ESKD in outpatient nephrology clinics. On the contrary, this evaluation is critical for several reasons. First, this is the reference of care for high-risk CKD patients. Second, in patients referred to nephrology ESKD overcomes mortality at variance with unselected CKD population [23,24]. Third, nephrologists prescribe RASI, potentially associated with high sK [22], to most patients in order to slow down CKD progression [22,23,24,25,26]. Fourth, in this specific setting a progressive decline of renal function, main determinant of sK increase, is more commonly observed with respect to other clinical settings [23,24]. Finally, novel K binders have been now developed to treat chronic hyperkalaemia and improve effectiveness of RASI therapy [21,22]; therefore, knowing the hyperkalaemia burden in renal clinics, where these new drugs will be largely used, becomes a preliminary but essential step. 

To fill this important gap of knowledge, we studied a large prospective cohort of ND-CKD patients under stable nephrology care to verify in outpatient CKD clinics (I) the prevalence of hyperkalaemia at referral and after 12 months, (II) the hyperkalaemia-related risk of ESKD and all-cause death in the subsequent period of follow up. To gain a proper analysis of the effect of sK on kidney survival, the effect of exposure was evaluated by competing risk analysis to take into account the potential occurrence of death before ESKD. Analyses were adjusted for major confounders, including RASI prescription and eGFR change in the first year of nephrology care.

## 2. Materials and Methods

This is a multicentre prospective study pooling data from four established cohorts (Figure 1). 

Cohorts included ND-CKD patients (eGFR < 60 mL/min/1.73 m^2^ or proteinuria >0.15 g/24 h) under stable care in Italian outpatients nephrology clinics to gain information on clinical features and outcome of referred CKD. Methodological details are described in the published papers [23,25,26,27] and here summarized in Appendix. To the aims of this study, we excluded duplicate patients, those with missing sK level as patients with only referral visit.

We grouped patients according to the presence of hyperkalaemia, defined as serum K ≥ 5.0 mEq/L in the two study visits: no hyperkalaemia in either visit (absent), hyperkalaemia only at visit 1 (resolving), or only at visit 2 (new onset), or at both visits (persistent). Similar categories were created for RASI prescription: both visits, only visit 1, only visit 2, neither visit. 

Endpoints of the study were ESKD, that is, start of chronic dialysis therapy (>30 days) or kidney transplantation and all-cause death before ESKD, as derived by national registries. For survival analyses, patients were followed from visit 2 up to ESKD, all-cause death, or 31 December 2015 and censored on the date of the last control visit.

### Statistics

Continuous variables are reported as either mean ± standard deviation (SD) or median and interquartile (IQR) range based on their distribution. Comparison among hyperkalaemia categories was assessed by ANOVA or Kruskal-Wallis test, while changes in each category across two visits were performed by paired Student’s t-test or paired Wilcoxon test. Categorical variables are reported as percentage and were analysed using a Chi-square test or McNemar test to evaluate changes across two visits. 

Follow-up for survival analyses started at visit 2 and median follow-up value was estimated by inverse Kaplan-Meier approach. Incidence rates of ESKD and death before ESKD were reported as number of events/person-time and 95% confidence interval (CI) calculated assuming a Poisson distribution. Risk of ESRD versus death in whole population was compared according to Kochar et al. [28].

Competing risk approach was used in the survival analysis because in ND-CKD, ESKD and death before ESKD are competing events, that is, occurrence of death prevents ESKD. The competing risk analysis must be considered when the absolute percentage of competing event is >10% [29]; in our population, in fact, death before ESRD occurred in 14.3%. 

In the univariate analysis we built and compared cumulative incidence curves among categories by using Gray test [30], while multivariable Fine and Gray model was used to estimate the sub-distribution hazard ratio (sHR) and 95% CI [31]. 

Models were stratified by cohort to consider potential differences in the basal risks across the four cohorts and were adjusted for the following potential risk factors of ESKD recorded at visit 2: age, gender, diabetes, body mass index (BMI), systolic blood pressure, cardiovascular (CV) disease history, haemoglobin, eGFR, 24-h proteinuria, categories of RASI, as well as the eGFR change between baseline (visit 1) and 12-month control (visit 2). 

To take into account the non-normal distribution of 24-h proteinuria and the non-linear association with outcomes, this covariate was log-transformed. Non-linear association of sK as continuous variable with outcomes was tested adding in the model the quadratic value of sK and retained if the coefficient was significant. 

A two-tailed *P* value < 0.05 was considered significant. Data were analysed using STATA version 14 (Stata Corp. College Station, TX, USA) and Cmprsk, CrrSC packages of R software 3.3.1 (R Foundation for Statistical Computing, Vienna, Austria).

## 3. Results

### 3.1. Baseline Period

The whole cohort was originally composed by 2813 patients referred to 46 outpatient CKD clinics in Italy. According to exclusion criteria, we studied 2443 patients (Figure 1). As compared with included patients, those excluded because of missed visit 2 were older and had more severe disease, while sK and use of RASI did not differ (Appendix A). 

Included patients still had a high-risk profile, as testified by the high prevalence of diabetes and CV disease, the low eGFR value and the high BMI and blood pressure levels (Table 1). 

Mean sK did not differ in the two visits (4.81 ± 0.62 and 4.80 ± 0.59 mEq/L, respectively). Prevalence of hyperkalemia was similar at visit 1 (39%, 95% CI 37–41) and visit 2 (37%, 95% CI 35–39%) (*p* = 0.266), and it was dependent on CKD stage (Figure 2). Severe hyperkalemia (sK ≥ 6 mEq/L) was rare (4% at visit 1 and 3% at visit 2), as hypokalemia (sK < 3.5 mEq/L < 1% in the two visits), while hyperkalemia was mild to moderate in the vast majority of cases (Figure 2). 

When considering sK in the two visits, we found that 32% patients changed the hyperkalemia status (Appendix A). New onset and persistent hyperkalemia categories accounted for 37% of population (15% and 22% respectively) while hyperkalemia was never detected in 46% and resolving in 17%.

As shown in Table 2, across hyperkalaemia categories, proteinuria and serum phosphate increased from absent to persistent (*p* < 0.001), while haemoglobin and eGFR decreased (*p* < 0.001). The higher proteinuria levels in new onset and persistent categories was probably due to higher prevalence of diabetic kidney disease (Table 1). At visit 1, the number of antihypertensive drugs was 2.15 ± 1.23 per patient and increased to 2.42 ± 1.34 at visit 2 (*p* < 0.001). RASI prescription in the hyperkalaemia categories is reported in Table 2; overall, 79% of patients were prescribed RASI, the vast majority of patients having this therapy at either visit while 11% were treated only at visit 1 and 9% only at visit 2. Use of dual RAS blockade slightly increased overall (from 8% to 10%, *p* < 0.001) from visit 1 to 2. Conversely, anti-aldosterone drugs were prescribed in <1% at either visit. The nephrologist intervention also led to increased prescription of statin (from 32% to 38%) and epoetin (from 13% to 18%) (*p* < 0.001 for both), while intervention aimed at controlling extracellular volume (diuretic administration and/or low salt diet-urinary Na excretion ≤100 mEq/24 h), was observed in 71% patients at visit 1 and decreased to 56% at visit 2 (*p* < 0.001), likely due to improved BP control (Table 2).

### 3.2. Survival Analysis

Survival analyses started after visit 2. During a median follow-up of 3.6 years (IQR 3.0–5.9), we registered 567 ESKD and 349 all-cause death before ESKD, with incidence rate per 100 patient/years being 6.4 (95% CI 5.8–6.9) and 3.9 (95% CI 3.5–4.3), respectively.

The unadjusted competing risk analysis showed that in the overall population the cumulative incidence of ESKD was markedly higher than mortality throughout the entire follow up period (Kochar test *p*-value < 0.001) (Figure 3). 

When examining the four groups, incidence of ESKD was similarly higher in new onset and persistent hyperkalaemia versus absent and resolving group, while mortality did not differ (Figure 4).

The multivariable survival analysis examining the independent role of sK as continuous variable showed that 1 mEq/L higher sK at visit 2 was associated with 20% higher risk of ESKD (HR 1.20, 95% CI 1.04–1.39, *p* = 0.014) with no effect on mortality (HR 0.94, 95% CI 0.76–1.17, *p* = 0.57). The association between sK and outcomes was linear as testified by the non-significant effect of quadratic sK on ESKD (*p* = 0.79) and death (*p* = 0.52). Similarly, the sK change (mEq/L) in the two visits significantly predicted ESKD (HR 1.24, 95% CI 1.05–1.46, *p* = 0.011) with a neutral effect on mortality (HR 0.95, 95% CI 0.75–1.20, *p* = 0.65). 

Table 3 illustrates the incidence and multi-adjusted risks of ESKD and death in the four hyperkalaemia categories. The crude incidence of ESKD progressively increased from absent to persistent hyperkalaemia while the association with mortality was less evident. At multivariable analyses, new onset and persistent hyperkalaemia were associated with significantly increased risk of ESKD (full multi-adjusted models are reported in Appendix A). Conversely, no association of hyperkalaemia with death emerged. Hazards did not change when also the causes of CKD were added to the survival models (sensitivity analysis).

Since underuse of RASI and persistent/new onset hyperkalaemia categories showed higher ESKD risk (Appendix A), we did an exploratory analysis evaluating the effect of combing hyperkalaemia and RASI categories on patient outcome (Table 4); this analysis suggested that, as compared to reference (absent or resolving hyperkalaemia combined with use of RASI), risk of ESKD significantly increases by 57% when new-onset/persistent hyperkalaemia associates with non-use or discontinuation after visit 1 of RASI. These results suggest a multiplicative effect of the two factors on ESKD risk. Conversely, no effect on mortality risk of combination was found.

At variance with sK measured at visit 2, we did not detect any association of sK measured at visit 1 with the subsequent incidence of ESKD (HR 1.04, 95% CI 0.90–1.20, *p* = 0.620). Also mortality risk related to sK at visit 1 was not significant (0.97, 95% CI 0.79–1.19, *p* = 0.750). These results persisted when adding the 370 patients initially excluded because of missed visit 2 (Appendix A).

## 4. Discussion

We here provide insights into the true burden of hyperkalaemia in CKD patients under continuous nephrology care. We add knowledge on this critical issue because we estimated prevalence of hyperkalaemia using two sK tests with 12-month interval in a large sample of referred patients and evaluated the association between hyperkalaemia and outcome by comprehensive survival analyses. We found that moderate hyperkalaemia, as defined over two visits with 12-month interval, is common in ND-CKD patients under nephrology care, with a substantial portion of this population showing new-onset or persistent hyperkalaemia over one year of observation. These patients showed higher risk of ESKD, with no risk excess of death, in the subsequent 3.6 years-follow up.

Gaining insights into prevalence and prognostic role of hyperkalaemia in the setting of tertiary nephrology care is important because nephrology clinics are the appropriate reference of care for ND-CKD patients, as testified by favourable global prognosis in referred patients as compared to those never or inconsistently followed by a nephrologist [23,32,33,34,35,36]. In this setting, improving risk stratification is mandatory to optimize practice of nephrology workforce, which is limited today and projected to further shrink in the next future [37]. Furthermore, at variance with unreferred CKD, where death overcomes ESKD, the natural fate of CKD under nephrology care is progression to ESKD [23,24,25,26,33,34,35,36,38,39,40]; in this regard, it is important to note that a recent survey among European nephrologists has disclosed that in CKD stage 5 the main driver to start renal substitutive therapy is the clinical picture, as currently recommended by international guidelines [41,42], with refractory hyperkalaemia eliciting immediate dialysis start by 100% of interviewed nephrologists [43]. Finally, hyperkalaemia is expected to be prevalent in this setting because nephrologists manage advanced CKD and, moreover, prescribe RASI to prevent CKD progression [1,2,3,4,5,6,7,16,17,18,19,41,44]. 

We found high prevalence of hyperkalaemia in our patient population, with high sK detected in as many as 54% patients in at least one of the two study visits. These figures are higher than previously reported [1,2,3,4,5,6,7,15]. Nevertheless, our study is hardly comparable with early work not only in terms of study setting but also for two features prevalent in renal clinics, that is, the low level of eGFR (77% had CKD stages 4,5) and the high rate of RASI use (more than two-thirds at either visit). This latter feature confirms the attitude of nephrologists toward maintenance of the only nephroprotective therapy today available in advanced CKD [23,25,45,46]. In particular, analysis of the NephroTest cohort has showed that prevalence of hyperkalaemia in ND-CKD was 6.5% [15]; however in that study patients were more healthy (younger age, lower comorbidities and minor renal dysfunction) as compared with our patients, and, in general with referred CKD patients. A likely reason is that additional and burdensome work-up in Physiology Department was required for enrolment into NephroTest.

The observed high prevalence of hyperkalaemia reinforces the need of answering to the critical question that remains unsolved so far, namely, as whether to what extent chronic hyperkalaemia increases the risk of dialysis initiation. The previous studies evaluating the renal risk related to hyperkalaemia have disclosed any or only weak association [1,4,10,13,14,15]. However, these earlier data were mostly attained in settings other than outpatient renal clinics, often including patients with preserved renal function and had short follow up. More important, no study has evaluated the ESKD risk by properly evaluating the competing risk of death and adjusted analyses for the confounding effect of eGFR decline.

We evaluated the effect of sK on ESKD risk by considering the competing risk of death before ESKD and the potential confounder of GFR decline. Serum K was consistently identified as independent risk factor of ESKD, when tested as continuous variable at visit 2 as well as sK change over the two visits. The latter finding anticipates the result attained when examining hyperkalaemia categories, where new-onset and persistent hyperkalaemia did portend a 30% higher risk of ESKD. Noteworthy, renal risk related to hyperkalaemia was independent also from the eGFR change between the two visits, thus excluding that it was merely dictated by the status of progressive CKD. Similarly, hyperkalaemia risk was also independent from older age and diabetes, that are main determinants of hyperkalaemia besides and beyond a low renal function [22]. 

Persistent or new-onset hyperkalaemia may also associate with lower use of RASI, that are the key nephroprotective agents today available for ND-CKD, therefore indirectly increasing ESKD risk. Although a cause-effect relationship can only be proven by a trial [46], it is interesting that a post-hoc analysis of RENAAL trial in diabetic CKD patients has shown the nephroprotective efficacy of losartan was in part offset by the hyperkalaemic effect of this drug [47]. Furthermore, a 8-week trial testing a new K binder in hyperkalaemic ND-CKD patients showed that more patients could continue the nephroprotective therapy with ARB in the K-binder arm (95%) than in the placebo arm (50%) [48]. Our exploratory analysis (Table 4) showed that the ESKD risk increases in hyperkalaemic patients not taking RASI on regular basis, thus suggesting that that the two factors should be examined jointly to a proper assessment of the hyperkalaemia-driven risk in ND-CKD. 

Interestingly, patient survival was not influenced by hyperkalaemia. Two potential reasons may explain this finding. First, in our referred patients, ESKD overcomes mortality, as expected in the tertiary nephrology care setting, and, second, nephrologists start dialysis in the presence of hyperkalaemia, refractory to therapy, to prevent deaths related to potential additional increases of sK.

Of note, main limitation to this study is that determinants of hyperkalaemia could not be assessed because profile of aldosterone/steroids, bicarbonate levels/supplementation and dietary K intake were not available in most patients; however, analysis of determinants was not the aim of this outcome study. On the other hand, strengths of the study are the sample size that is relatively large considering the setting of tertiary nephrology care, the evaluation of ESKD risk that took into account the potential confounder of death before ESKD and the adjustment for the extent of eGFR change that allows to dissect the role of hyperkalaemia from that of GFR decline on ESKD onset.

## 5. Conclusions

In conclusion, our study provides evidence that in ND-CKD patients under stable nephrology care, true mild-to-moderate hyperkalaemia (sK 5.0–6.0 mEq/L) is (I) common with 37% patients showing new-onset or persistent hyperkalaemia over two visits with 12-month interval and (II) portends *per se* a 30% higher risk of ESKD, that is independent from the rate of eGFR decline, while not affecting patient survival.

## Figures and Tables

**Figure 1 jcm-07-00499-f001:**
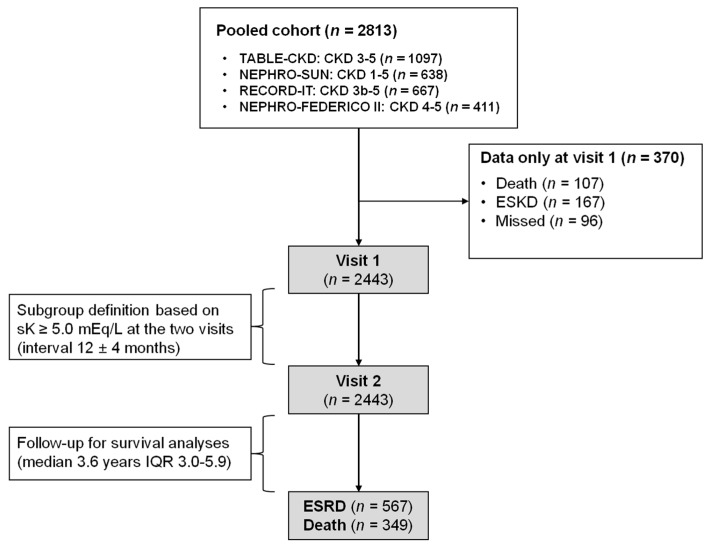
Study flow chart.

**Figure 2 jcm-07-00499-f002:**
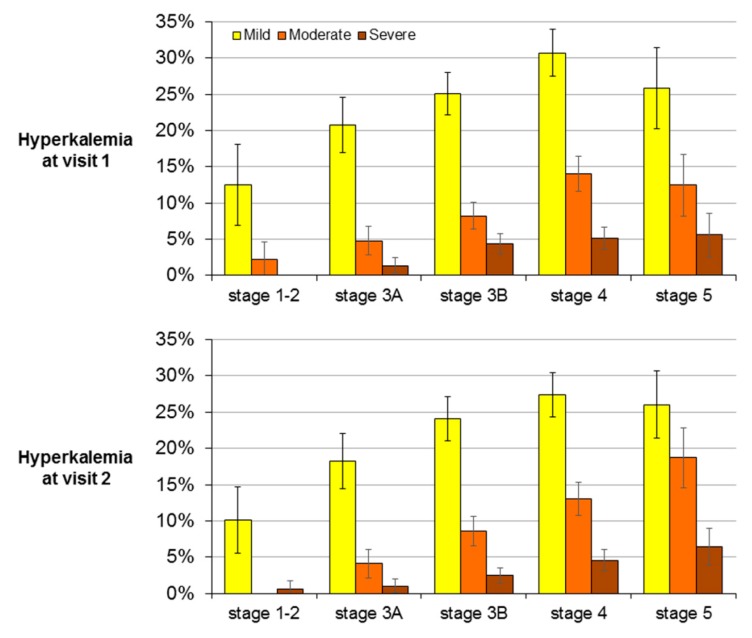
Prevalence (% and 95% CI) of mild, moderate and severe hyperkalaemia by CKD stage at the two study visits. Mild: serum K 5.0–5.4 mEq/L; Moderate: 5.5–5.9 mEq/L; Severe: ≥6.0 mEq/L.

**Figure 3 jcm-07-00499-f003:**
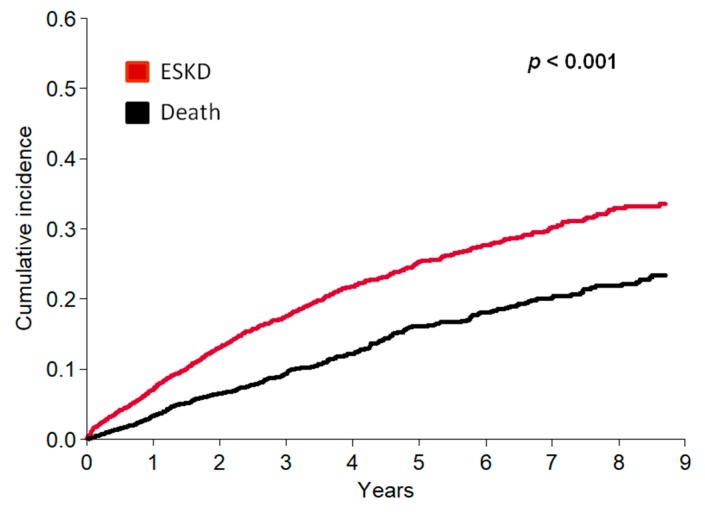
Cumulative incidence of ESKD and death-before-ESKD in the whole study population.

**Figure 4 jcm-07-00499-f004:**
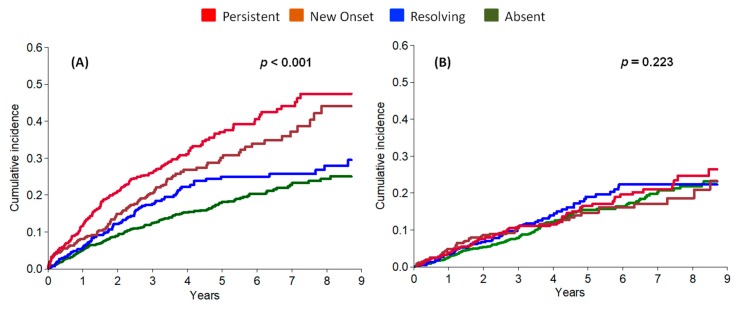
Cumulative incidence after visit 2 of ESKD (**A**) and all-cause death before ESKD (**B**), by competing risk analysis, by hyperkalaemia category.

**Table 1 jcm-07-00499-t001:** Demographics of patients at Visit 1.

		Hyperkalaemia				*p*
	Overall	Absent (No-No)	Resolving (Yes-No)	New Onset (No-Yes)	Persistent (Yes-Yes)	
Number (%)	2443 (100)	1121 (46)	415 (17)	363 (15)	544 (22)	
Age (years)	65.1 ± 14.6	64.4 ± 15.2	66.6 ± 13.6	63.6 ± 15.4	66.4 ± 13.4	0.01
Males (%)	58	57	58	64	59	0.12
BMI (kg/m^2^)	27.8 ± 5.1	27.7 ± 5.0	27.8 ± 5.1	27.5 ± 4.8	28.1 ± 5.4	0.35
Diabetes (%)	28	26	28	29	31	0.15
CV disease (%)	36	35	37	36	36	0.86
Smoking (%)	13	13	15	13	12	0.68
eGFR (mL/min/1.73 m^2^)	35.0 ± 17.3	39.0 ± 19.3	33.1 ± 15.0	34.3 ± 16.7	28.6 ± 12.3	<0.001
Systolic BP (mmHg)	139 ± 20	139 ± 20	139 ± 20	139 ± 21	141 ± 19	0.281
Renal disease (%)						<0.001
HTN	30	33	32	28	24	
DKD	13	10	13	16	17	
GN	17	19	15	15	15	
TIN	9	10	9	10	8	
PKD	5	5	5	5	6	
Other	7	7	6	8	9	
Unknow	19	17	20	19	22	

BMI, body mass index; CV, cardiovascular; BP, blood pressure; HTN, hypertensive nephropathy; DKD, diabetic kidney disease; GN, glomerulonephritis; TIN, tubulo interstitial nephropathy; PKD, polycystic kidney disease. Continuous variables are reported as mean ± SD

**Table 2 jcm-07-00499-t002:** Clinical characteristics of patients by hyperkalaemia status (sK ≥ 5.0 mEq/L) at visit 1 and 2.

	Absent (No-No) (*n* = 1121)		Resolving (Yes-No) (*n* = 415)		New Onset (No-Yes) (*n* = 363)		Persistent (Yes-Yes) (*n* = 544)	
	Visit 1	Visit 2	Visit 1	Visit 2	Visit 1	Visit 2	Visit 1	Visit 2
Systolic BP (mmHg)	139 ± 20	135 ± 19 *	139 ± 20	136 ± 19 *	139 ± 21	134 ± 19 *	141 ± 19	137 ± 18 *
Diastolic BP (mmHg)	80 ± 11	78 ± 11 *	79 ± 11	77 ± 10 *	80 ± 11	77 ± 10 *	79 ± 11	77 ± 10 *
Potassium (mEq/L)	4.38 ± 0.40	4.40 ± 0.37	5.32 ± 0.35	4.55 ± 0.31	4.57 ± 0.32	5.30 ± 0.30	5.47 ± 0.44	5.46 ± 0.42
Glucose (mg/dL)	104.4 ± 33.4	103.9 ± 32.3	112.6 ± 48.6	110.9 ± 39.6	106.7 ± 36.9	108.6 ± 43.5	108.9 ± 42.9	107.7 ± 37.4
Phosphate (mg/dL) ^	3.67 ± 0.78	3.71 ± 1.11	3.84 ± 0.75	3.78 ± 0.77	3.77 ± 0.74	3.84 ± 0.95	3.95 ± 0.79	4.02 ± 0.88 *
Haemoglobin (g/dL) ^	13.0 ± 1.8	12.9 ± 1.7 *	12.7 ± 1.8	12.7 ± 1.7	12.5 ± 1.7	12.5 ± 1.7	12.2 ± 1.7	12.2 ± 1.6
eGFR (mL/min/1.73 m^2^) ^	39.0 ± 19.3	38.3 ± 20.3 *	33.1 ± 15.0	33.6 ± 17.1	34.3 ± 16.7	31.4 ± 16.6 *	28.6 ± 12.3	25.8 ± 12.3 *
eGFR change (mL/min/year) ^	−0.55 ± 10.66		1.13 ± 10.34		−2.93 ± 9.63		−2.90 ± 7.81	
Proteinuria (g/24 h) ^	0.30 (0.12–1.00)	0.28 (0.11–0.90) *	0.36 (0.12–1.10)	0.40 (0.12–1.06)	0.53 (0.15–1.60)	0.52 (0.14–1.49)	0.60 (0.18–1.50)	0.68 (0.19–1.45)
RASI °								
None (%)	33	34	28	33	33	31	23	30
CEI or ARB (%)	60	55	65	60	58	57	70	60
Dual blockade (%)	7	11	7	7	10	13	7	10

Data are mean ± SD or median (IQR) or percentage of patients. BP, blood pressure; eGFR, estimated glomerular filtration rate by EPI equation; RASI, renin angiotensin system inhibitors; CEI, converting enzyme inhibitor; ARB, angiotensin II receptor blocker. ^ *p* < 0.05 for visit 1 and visit 2 among the groups. * *p* < 0.05 vs. Visit 1. ° RASI distribution differs between visit 1 and 2 for Absent and Persistent group.

**Table 3 jcm-07-00499-t003:** Incidence and multi-adjusted risk of ESKD and all-cause death by hyperkalaemia status.

Hyperkalaemia	ESKD			All-Cause Death		
	Incidence (Events/Pts)	Incidence Rate *per 100-pt-y* (95%-CI)	sHR (95%-CI)	Incidence (Events/Patients)	Incidence Rate *per 100-pt-y* (95%-CI)	sHR (95%-CI)
Absent	188/1121	4.32 (3.72–4.98)	Reference	147/1121	3.38 (2.85–3.97)	Reference
Resolving	93/415	5.98 (4.83–7.33)	0.98 (0.72–1.33)	65/415	4.18 (3.23–5.33)	1.01 (0.73–1.36)
New-onset	105/363	8.23 (6.73–9.96)	1.34 (1.05–1.72)	51/363	4.00 (2.98–5.26)	0.94 (0.64–1.38)
Persistent	181/544	10.52 (9.04–12.17)	1.27 (1.02–1.58)	86/544	5.00 (3.99–6.17)	0.91 (0.67–1.25)

sHR, sub-hazard ratio; CI, confidence interval. Bold indicates statistical significance. Hyperkalaemia is defined by sK ≥ 5.0 mEq/L over the two study visits. Fine and Gray models are stratified by cohort and adjusted for variables at visit 2 (age, gender, diabetes, CV disease, BMI, haemoglobin, eGFR, 24 h proteinuria, systolic BP), RASI therapy over the two visits and for eGFR change between the two study visits. The full model is shown in the Appendix (Appendix A-Appendix).

**Table 4 jcm-07-00499-t004:** Incidence and adjusted risk of ESKD in patients stratified in subgroups by the combination of hyperkalaemia and RASI categories over the two study visits.

Hyperkalaemia	RASI	Incidence (events/pts)	Incidence Rate (per 100-pts-y)	sHR (95% CI)	*p*
Absent/Resolving	Yes	137/1.018	3.18 (2.67–3.75)	Reference	-
Absent/Resolving	No	144/518	9.02 (7.61–10.62)	1.09 (0.83–1.44)	0.540
New onset/Persistent	Yes	171/632	7.43 (6.36–8.63)	1.18 (0.94–1.49)	0.140
New onset/Persistent	No	115/275	16.55 (13.67–19.87)	1.57 (1.20–2.06)	0.001

Hyperkalaemia defined by sK ≥ 5.0 mEq/L over the two study visits. sHR, sub-hazard ratio; CI, confidence interval; RASI, inhibitors of renin-angiotensin system. Yes, RASI prescription at both visits or only visit 2; No, RASI prescription at only visit 1 or at none of the two visits. The model is stratified by cohort and adjusted for variables at visit 2 (age, gender, diabetes, CV disease, BMI, haemoglobin, eGFR, 24 h proteinuria, systolic BP) as for eGFR change between the two study visit.

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
