# Peer review of "Competing-Risk Analysis of Death and End Stage Kidney Disease by Hyperkalaemia Status in Non-Dialysis Chronic Kidney Disease Patients Receiving Stable Nephrology Care"

_jcm, 2018, doi:10.3390/jcm7120499_

Reviewer 1 Report

The present study by Provenzano M et al examined hyperkalemia burden in non-dialysis CKD patients in outpatient’s settings in Italian population.  It is multicenter study 46 nephrology clinics and about 2443 patients were participated. Patients are divided based on hyperkalemia status such as absent, resolving, new-onset, and persistent. Author emphasized  that competing risk approach of death before ESKD and adjustment in severity of CKD progression as GFR declines. Overall, this study provides evidence that in ND-CKD has mild to moderate hyperkalemia is relatively common but not related to the mortality. It is interesting study with more patients and definitely add more information for nephrologist to handle sK level in ND-CKD patients.

i)               Among all clinical characterization of patients, why did not author measured aldosterone concentration and other steroid profile? It would be interesting to see if aldosterone goes up when sK level increases in those patients.

ii)              Introduction should be more clear. Also include strong background from De Nicola L, et al J Nep 2018 study as it explain the purpose of the current study

iii)             It is not novel study as author claimed to be the first such a study. But still it is useful study for the prediction.  Sentences should be softened.

iv)             Introduction: change properly to no detailed study has evaluated ….

v)              Use same acronym throughout the manuscript. RASI or RAASI

vi)             Why this study results cannot be applied for people other than Caucasians? If there is previous about race differences, please quote.

Author Response

Thank you for your comments.

Response is uploaded

Reviewer 2 Report

This is an interesting study on patients with chronic kidney disease, in whom risk factors of kidney failure progression and death are still not sufficiently explored. I have few questions concerning presented study:

1) in table 2 the Authors present that patients with new onset hyperkalemia (yes-yes group) during second visit had higher potassium level, lower GFR and more patients were taking dual RAS blockers, similar observations were made in the persistent group (yes-yes). Could hyperkalemia and GFR decrease be caused by higher RAAS inhibitors intake? (especially double blockade?) Why more patients were taking RAAS inhibitors?

2) how authors will explain that in the new onset group (no-yes) and persistent group (yes-yes) patients from the beginning of the study had higher proteinuria? Also in the persistent group seems that patients had lower GFR. Such patients could have more advanced kidney failure and concomittant disorders, causes and time of kidney failure diagnosis should be compared. If patients with higher kidney failure stage + hyperkalemia are compared to patients with lower kidney function decline showed results might be not totally true.

3) patient's diet (especially food supplements) are crucial in hyperkalemia management and should be taken under consideration in future studies.

Author Response

Thank you for your comments.

Response letter is uploaded 
